# Mitochondria and Pharmacologic Cardiac Conditioning—At the Heart of Ischemic Injury

**DOI:** 10.3390/ijms22063224

**Published:** 2021-03-22

**Authors:** Christopher Lotz, Johannes Herrmann, Quirin Notz, Patrick Meybohm, Franz Kehl

**Affiliations:** 1Department of Anesthesiology, Intensive Care, Emergency and Pain Medicine, University Hospital of Wuerzburg, 97080 Wuerzburg, Germany; Lotz_C@ukw.de (C.L.); Herrmann_J4@ukw.de (J.H.); Notz_Q@ukw.de (Q.N.); Meybohm_P@ukw.de (P.M.); 2Department of Anesthesiology and Intensive Care, Karlsruhe Municipal Hospital, 76133 Karlsruhe, Germany

**Keywords:** cardioprotection, preconditioning, ischemia/reperfusion injury, volatile anesthetics

## Abstract

Pharmacologic cardiac conditioning increases the intrinsic resistance against ischemia and reperfusion (I/R) injury. The cardiac conditioning response is mediated via complex signaling networks. These networks have been an intriguing research field for decades, largely advancing our knowledge on cardiac signaling beyond the conditioning response. The centerpieces of this system are the mitochondria, a dynamic organelle, almost acting as a cell within the cell. Mitochondria comprise a plethora of functions at the crossroads of cell death or survival. These include the maintenance of aerobic ATP production and redox signaling, closely entwined with mitochondrial calcium handling and mitochondrial permeability transition. Moreover, mitochondria host pathways of programmed cell death impact the inflammatory response and contain their own mechanisms of fusion and fission (division). These act as quality control mechanisms in cellular ageing, release of pro-apoptotic factors and mitophagy. Furthermore, recently identified mechanisms of mitochondrial regeneration can increase the capacity for oxidative phosphorylation, decrease oxidative stress and might help to beneficially impact myocardial remodeling, as well as invigorate the heart against subsequent ischemic insults. The current review highlights different pathways and unresolved questions surrounding mitochondria in myocardial I/R injury and pharmacological cardiac conditioning.

## 1. Introduction

Myocardial ischemic injury is composed of two different entities with distinct molecular mechanisms. The first is ischemia itself, and the second is reperfusion injury. Ischemia damages myocardial tissue and renders it susceptible to subsequent injury by reperfusion as a second hit. Myocardial reperfusion is responsible for up to 50% of the overall damage of combined ischemia/reperfusion (I/R) injury. Nevertheless, in case ischemia is not terminated tissue damage becomes irreversible and as such reperfusion of myocardial tissue is indispensable [1].

For decades, research has focused on mechanisms of myocardial I/R injury and on how to minimize the resulting myocardial infarct size by increasing the intrinsic resistance of the heart. Research efforts were sparked by the discovery of ischemic preconditioning by Murray et al. in 1986. Short periods of ischemia interspersed by reperfusion followed by a sustained ischemia limited the resulting infarct size to 25% compared to the control group [2]. Hopes of clinically applicable benefits increased shortly thereafter, as a number of drugs were shown to exert similar conditioning effects. These, amongst others, include opioids [3] and volatile anesthetics. Cardioprotective effects of volatile anesthetics have been known for three decades and cardiac protection is similar in magnitude to ischemic preconditioning [4]. Furthermore, different time frames and modalities have been discovered, such as delayed anesthetic preconditioning and postconditioning or remote ischemic preconditioning (RIPC) [5,6]. Opioids, on the other hand, are an example of conditioning agents whose receptors concomitantly act as molecular targets for other cardioprotective triggers [7,8]. Subsequent research efforts identified an almost uncountable number of molecular pathways and interactions triggering and mediating this intrinsic invigorating response. The findings helped to draw a picture of complex intracellular networks and interactions within cardiac myocytes, advancing the field of cardiac research beyond cardiac conditioning or even the heart as many organs share similar mechanisms to protect themselves against ischemic damage. Moreover, the communality of molecular mechanisms between different pharmacologic conditioning agents is overwhelming, emphasizing the presence of endogenous defense mechanisms not unique to the heart, but a superordinate system.

The centerpieces of this system are the mitochondria. The myocardium is a particular mitochondria rich tissue with an aerobic ATP production of approximately 6 kg per day [9]. Each cardiac myocyte contains about 5000 mitochondria, comprising 40% of the cytoplasmic space. Mitochondria originate from bacterial endosymbiosis of primitive forms of eukaryotic cells and comprise their own mitochondrial (mt) DNA. Mitochondrial DNA is double-stranded and codes for 13 proteins of the mitochondrial respiratory chain [10]. However, the majority of all mitochondrial >1000 proteins [11] is translated from nuclear DNA. Although mitochondria are best known for their role as a cellular energy source, they represent an extremely complex and dynamic organelle, almost acting as a cell within the cell. Mitochondria comprise a plethora of functions at the crossroads of cell death or survival. These include the maintenance of aerobic ATP production, redox signaling, calcium transport, and pathways of cell death. The current review focuses on the different pathways and unresolved questions surrounding mitochondria in myocardial I/R injury and pharmacological cardioprotection.

## 2. Mechanisms of Myocardial Damage and Cell Death after Ischemia/Reperfusion

Myocardial ischemia is characterized by profound intracellular metabolic alterations. Hypoxia forces a switch from aerobic to anaerobic metabolism and accumulating H^+^ ions activate the Na^+^/H^+^ ion exchanger. Na^+^/K^+^ ATPase activity is reduced by concomitant ATP depletion. The resulting Na^+^ accumulation leads to reversed mode operation of the Na^+^/Ca^2+^ ion exchanger, culminating in intracellular and mitochondrial Ca^2+^ overload [12,13]. These mechanisms are exaggerated during the beginning of myocardial reperfusion due to damage to the plasma and sarcoplasmic membranes, rapid washout of lactate, restoration of intracellular pH, and mitochondrial respiratory chain dysfunction. The reintroduction of O_2_ onto a dysfunctional respiratory chain leads to the production of massive amounts of reactive oxygen species (ROS) and further exaggeration of intracellular Ca^2+^ levels. Ca^2+^ enters the mitochondria via the mitochondrial Ca^2+^ uniporter (mCU) [14] and promotes the opening of the mitochondrial permeability transition pore (mPTP). The mCU resides in close proximity to the respiratory chain and is driven by the electrochemical gradient across the inner mitochondrial membrane generated by the electron transport chain [15]. The channel becomes active through the rapid restoration of the mitochondrial membrane potential during reperfusion, and it has been shown that knockout mCU^−/−^ mice are protected against I/R injury. However, the channel itself likely is not a therapeutic target as these mice also lack contractile responsiveness to acute β-adrenergic receptor stimulation and have a reduced bioenergetic reserve [16]. On the other hand, MCUB, a constituent incorporated into the mCU under stress conditions, might be a more promising target. MCUB regulates the formation and stoichiometry of mCU subunits and its overexpression limits mitochondrial Ca^2+^ uptake during I/R injury. While also impairing mitochondrial bioenergetics, Ca^2+^ independent mechanisms might compensate these energy deficits during chronic overexpression of MCUB. This could limit cell death during the later stages of I/R injury and heart failure development [17].

## 3. Intracellular Pathways of Pharmacologic Cardiac Conditioning

Traditionally, mechanisms of cardiac conditioning were divided into triggers of the protective response acting at the cell surface and downstream mediators targeting intracellular effectors. Many of these molecular pathways converge at the level of the mitochondrion. Considering the extent of published data (a total of >11,000 hits on pubmed utilizing the term “cardiac conditioning”), it is nearly impossible to provide a comprehensive overview on all interacting pathways. However, major building blocks of pre-mitochondrial and mitochondrial cardioprotective signaling can be identified (Figure 1). Cardioprotective signaling triggered by pharmacologic agents commences via G-protein and non-G-protein coupled receptor pathways with e.g., β-adrenergic receptors [18,19] activating Gαs, proteinkinase A, subsequently increasing nitric oxide production via endothelial nitric oxide synthase [20,21]. This results in proteinkinase G and proteinkinase C (PKC) activation [22]. PKCε links cytoplasmatic and mitochondrial signaling influencing ROS production, mPTP opening and the mitochondrial ATP-dependent K^+^ channel [23,24]. PKC signaling is also linked to the reperfusion injury salvage kinase (RISK) pathway subsequent to the activation of inhibitory G proteins [25,26]. The RISK pathway is considered the main pro-survival kinase cascade with toll-like-receptors (TLR) and phosphatidylinositol 3-kinase (PI3K)/Akt signaling preventing pro-inflammatory events and apoptosis through cross-talking with nuclear factor (NF)-κB [27,28], as well as glycogen synthase kinase 3β (GSK3β). GSK3β transfers the protective signal downstream to targets that act at or in proximity to the mPTP [29], such as Bcl-2 family proteins, ANT, VDAC, cyclophilin D, and hexokinase II [30]. Another major survival pathway is the survival factor enhancement (SAFE) pathway consisting of Janus kinase (JAK)/signal transducer and activator of transcription (STAT) signaling after activation at the transcriptional level by cytokines, i.e., interleukin 6. STAT3 is primarily a transcription factor initiating the transcription of cardioprotective and anti-apoptotic proteins [31]. It is still questionable if STAT3 elicits direct mitochondrial actions, including the regulation of ROS production by regulating the activities of the electron transport chain. STAT3 is believed to associate with respiratory complex I and enhance complex I activity [32]. It also binds to cyclophilin D (CypD) inhibiting the opening of the mPTP [33,34,35].

All signaling pathways interact in terms of a cardioprotective gene and pathway network with temporo-spatially differences and likely remote conditioning of non-ischemic areas. In this regard, ischemic conditioning has recently been suggested to induce early activation of Ca-, adipocytokine, and insulin signaling with a central role for neprilysin and the STAT family in intrinsic remote conditioning. This response could prevent adverse myocardial remodeling on top of limiting acute cell death [36].

## 4. Mitochondrial Respiratory Chain and ATP Production

Cardiac mitochondria can be subdivided into two distinct populations: subsarcolemmal mitochondria (SSM) and interfibrillar mitochondria (IFM) [37]. SSM are more susceptible to Ca^2+^ overload, release of cytochrome c [38], and a decreased rate of oxidative phosphorylation [39]. Ischemic damage occurs in a progressive, time-dependent manner and during prolonged periods of ischemia (>60 min) damage to all constituents of the respiratory chain can be expected. It is important to note that mitochondrial damage mainly occurs during myocardial ischemia via compromised respiration and ROS production. Mitochondria mediated cardiac myocyte injury primarily happens during reperfusion. Reperfusion exaggerates oxidative stress and Ca^2+^ overload as respiratory chain dysfunction leads to the production of high levels of reactive oxygen species after reprovisioning O_2_. A total of up to 40% of I/R injury occurs during reperfusion.

Respiratory complexes I and III are the primary sites of ROS production, while complex II (although resisting ischemic damage for a longer time (45–60 min)) contributes to ROS production with reverse electron flow to complex I. Reverse electron flow depends on the mitochondrial membrane potential (Δψ_m_) and is facilitated in ischemic hyperpolarized non-phosphorylating mitochondria [40]. Dysfunction of complex IV is accompanied by a loss of cardiolipin, a phospholipid of the inner mitochondrial membrane facilitating electron channeling via the association of complexes III and IV into supercomplexes [41].

Studies of pharmacological conditioning have shown that cardioprotection alters a number of mitochondrial proteins with most changes occurring in complexes of the electron transport chain [42]. A total of 26 potential phosphorylation sites of isoflurane-induced cardioprotection were be identified within 19 mitochondrial proteins, including those directly related to oxidative phosphorylation [43]. Isoflurane and sevoflurane directly inhibit respiratory complex I [44] as a beneficial mechanism both during ischemia and reperfusion. Furthermore, isoflurane elicited a partial dissipation of the mitochondrial membrane potential and mild uncoupling reducing mitochondrial Ca^2+^ uptake [45] and ROS [46]. It has been shown that the reversible blockade of electron transport during ischemia attenuates the production of ROS during ischemia and reperfusion [47], a finding confirmed for volatile anesthetics early and late during reperfusion [48]. Preconditioning with isoflurane preserved the activity of complex III and stabilized the formation of respiratory supercomplexes III/IV, findings accompanied by a diminished mitochondrial susceptibility to Ca^2+^-induced swelling [49]. Moreover, direct actions of volatile anesthetics onto respiratory complexes might lead to beneficial effects independent of a pre- or postconditioning protocol. Mitochondria from animals receiving sevoflurane anesthesia had a preserved mitochondrial respiratory control ratio and unimpeded activities of respiratory complex I and complex IV after I/R during myocardial injury without the use of a specific conditioning protocol. In this regard, a significantly higher portion of complex I was found to be in its inactive/dormant form [50]. Deactivation of complex I could be an important mechanism preventing reverse electron flux and massive ROS production [51]. Moreover, reversible inhibition of the electron transport chain at the beginning of reperfusion with amobarbital reduced the extent of myocardial damage [52,53]. It is important to note that a small amount of “signaling ROS” is considered essential to both trigger and mediate pharmacological infarct size reduction. “Signaling ROS” originate from complex III [54] and likely nullify the therapeutic usefulness of ROS scavengers [55].

A proposed modulator of the respiratory chain is signal transducer and activator of transcription 3 (STAT3). STAT3 translocates to mitochondria after phosphorylation of Ser^727^. In cells exposed to hypoxia/reoxygenation zinc decreased ROS production and suppressed succinate dehydrogenase (SDH) activity via STAT3 phosphorylation at the onset of reperfusion [56]. Transgenic mice overexpressing a mitochondria-targeted STAT3 exhibit a persistent, partial blockade of electron transfer through complex I, a genotype associated with decreased myocardial infarct size, and increased survival after I/R injury [57]. Mitochondrial actions of STAT3 are on top of its actions as a transcription factor (regulating antioxidative, antiapoptotic, and proangiogenic gene expression). As such, STAT3 could be considered a central mediator of cardioprotection and concomitantly represents the complexity of multifunctionality and interactions of cardioprotective signaling networks (Figure 2). However, the mitochondrial presence and actions of STAT3 are in doubt as recent results found STAT3 solely along the T-tubules and in the nucleus without any effects of hypoxia or hypoxia/reoxygenation. Moreover, pSTAT3^Ser727^ and total STAT3 could not be detected in pure mitochondrial preparations [58].

Essentially, numerous functions of the respiratory chain and mitochondrial metabolism are connected to mitochondrial Ca^2+^ handling. Entry of Ca^2+^ into the mitochondrial matrix is governed by the mitochondrial Ca^2+^ uniporter (mCU). The mCU is a molecular complex regulated by the essential mCU regulator (EMRE) linking mCU with mitochondrial calcium uptake 1 (MICU1). MICU1 itself is connected to MICU2. The latter two have been suggested as gatekeepers, although the regulatory mechanism of the mCU is not fully understood. A current model suggests a sigmoidal response, whereas at low Ca^2+^ concentrations, MICU1 and MICU2 prevent calcium entry. Higher intermembrane Ca^2+^ concentrations cause a conformational change with the entry of large amounts of Ca^2^ [59]. Mitochondrial Ca^2+^ flux in turn impacts pyruvate dehydrogenase, complex V and the tricarboxylic acid cycle, as well as handling of Na^+^ and H^+^. Increased mitochondrial Ca^2+^ diverts protons away from the ATP synthase via activation of the Na^+^/Ca^2+^ and Na^+^/H^+^ exchanger. This potentially uncouples respiration from ATP production as complex V switches to reverse mode operation to extrude H^+^ [60].

## 5. Mitochondrial Mechanisms of Cell Death

Cell death is the irreversible degeneration of vital cellular functions with a concomitant disintegration of cellular membranes. A number of molecular mechanisms can be distinguished including regulated or programmed cell death, overall leading to an apoptotic or necrotic phenotype. In 2018, the Nomenclature Committee on Cell Death (NCCD) defined guidelines for the definition and interpretation of different types of cell death [61]. Besides apoptosis and necrosis, pyroptosis, and ferroptosis (resulting from lipid peroxidation after mitochondrial ROS release), have been described as mitochondria dependent mechanisms of cell death (Figure 3). Opening of the mitochondrial permeability transition pore (mPTP) is considered a hallmark at the crossroads of cell death and survival, whereas many cardioprotective pathways converge to inhibit its opening.

The mPTP acts as a non-selective channel for ions and solutes up to 1.5 kDa across the inner mitochondrial membrane. Opening is not only elicited by increased Ca^2+^ and ROS, but modulated via cyclophilin D, free fatty acids, or a diminished transmembrane potential. The molecular identity—as well as the gating mechanisms of the mPTP—are still controversial. Evidence suggests that the mPTP is actually formed by the F_1_F_0_-ATPase, i.e., complex V of the respiratory chain [62]. It has been proposed that the C-terminal subunit e of the F_1_F_0_-ATPase causes conformational changes transforming the c-ring into an ion conductive pore as a “death finger” [63]. In consequence mitochondrial swelling, membrane depolarization, uncoupling of oxidative phosphorylation, and ATP depletion leads to cell death [64]. Moreover, if the F_1_F_0_-ATPase is the mPTP, this represents a direct link of mitochondrial respiratory chain function and cell death. Hence, the passive, straightforward, mechanism of preserving mitochondrial respiratory chain function likely represents the most important step in inhibiting mPTP opening. However, the view that the F_1_F_0_-ATPase is the pore forming component of the mPTP has been challenged as the absence of the c-subunit did not abrogate mitochondrial permeability transition [65].

Inhibition of mPTP opening has been a target of pharmacologic cardioprotection for a long time [66], although some of the earlier findings might need to be revisited in light of progress on the identity of the mPTP. Volatile anesthetics [67], noble gases [68], and opioids [69,70] have been associated with mPTP inhibition during pre- and postconditioning, whereas the activation of the RISK pathway and downstream activation of eNOS, GSK-3β, Hexokinase II, PKC-ε, and the mitochondrial ATP-dependent potassium channel (mitoK_ATP_) are involved [71]. Desflurane exposure promotes mitochondrial resistance to Ca^2+^ induced mitochondrial Ca^2+^ release [72], which is linked to the mitochondrial translocation of PKCε [73]. STAT3 knockout mice tolerated less Ca^2+^ until mPTP opening [74]. Moreover, cyclosporin A is a known inhibitor of mPTP opening [75] targeting cyclophilin D, which regulates mPTP activity [76].

### 5.1. Necrosis

The NCCD guidelines include mitochondrial permeability transition-driven necrosis and necroptosis. As the mPTP plays a major role in myocardial infarction and cardiac conditioning, this type of cell death can be considered the major mechanism of cell destruction subsequent to the disruption of respiratory chain functionality. Hence, opening of the mPTP is directly linked to cell death in metabolically defunct and energy deprived cardiac myocytes.

Necroptosis is a regulated form of necrotic cell death utilizing defined signaling pathways. Necroptosis is triggered via the activation of death receptors, often linked to inflammatory mediators, such as TNF-α or interleukines. Tumor necrosis factor receptor 1 (TNFR1) and subsequent signaling via receptor-interacting protein 1 (RIP1) mediates the activation of receptor-interacting protein 3 (RIP3) and mixed lineage kinase domain-like (MLKL). These altogether facilitate disruption of the cell membrane. Alternative activation of calcium-calmodulin-kinase II (CaMKII) by the RIP3/RIP1/MLKL/Fas-associated protein with death domain (FADD) complex induces mitochondrial dysfunction and membrane permeabilization via cyclophilin D, voltage-dependent anion-selective channel (VDAC), and adenine nucleotide translocator (ANT) [77,78]. The participation of necroptosis in myocardial cardioprotection was shown in perfused guinea pig hearts utilizing necrostatin-1 as necroptosis inhibitor [79]. Moreover, melatonin ameliorated TNF-α levels and suppressed RIP3-MLKL/CaMKII signaling when given just prior to myocardial reperfusion [80]. Ischemic preconditioning inhibited the formation of MLKL oligomers and their subsequent translocation within the plasma membrane, although no upstream changes in RIP1/RIP3 expression or RIP3 phosphorylation could be found [81]. RIP1 and RIP3 also seem to mediate myocardial remodeling. Necrostatin-1 inhibited RIP1-dependent necrosis in murine hearts, leading to infarct size reduction and preservation of cardiac function in magnetic resonance imaging (MRI) scans 28 days after I/R injury [82]. In a large animal model of I/R injury, necrostatin-1 administration prior to reperfusion significantly reduced ischemic injury and preserved left ventricular function [83]. RIP3 knockout mice had a better ejection fraction and less hypertrophy in MRI scans 30 days after myocardial I/R injury [84]. All of these findings are likely related to a diminished inflammatory response as recently shown for dexmedetomidine induced preconditioning [85,86].

### 5.2. Apoptosis

Apoptosis only contributes a small amount of myocardial damage in the acute phase of myocardial I/R injury. In human autopsies, approximately 1–12% of cardiac myocytes were apoptotic in the border zone and only 0.04% in central infarction areas [87,88]. The number of apoptotic cells likely increases within the first 48 h and the detrimental impact of apoptosis [89] is more pronounced in the subacute phase of myocardial infarction. A four-fold increased rate of apoptosis was shown in patients dying from early symptomatic post-infarction heart failure due to left ventricular remodeling [90].

Molecular mechanisms of apoptosis are classified into death-receptor mediated (extrinsic) apoptosis and mitochondria mediated (intrinsic) apoptosis. Death receptor mediated apoptosis is activated after, e.g., the binding of apoptosis-stimulating fragment (Fas), TNF-α, or TNF-related apoptosis stimulating ligand (TRAIL). Death receptor activation leads to caspase-8 activation via complexes including RIPK1, FADD, and TRADD, followed by the cleavage and activation of caspases-3 and -7. This results in nuclear as well as membrane apoptotic changes. Hence, to some degree death receptor signaling overlaps with necroptosis signaling, although with different downstream effectors. Moreover, the activation of caspases 3 and 7 is shared by extrinsic and intrinsic apoptosis as common end-effectors. The intrinsic, mitochondria-mediated pathway is triggered by a number of stimuli including hypoxia/reoxygenation, ischemia/reperfusion, oxidative stress, nitrosative stress, proteotoxic stress, DNA damage, and increased Ca^2+^ concentration. These stimuli trigger activator BH3 and downstream Bcl-2-interacting domain death agonist (BID), Bcl-2-interacting mediator of cell death (BIM), as well as p53 upregulated modulator of apoptosis (PUMA), which conformationally activate Bcl-2-associated X protein (Bax) and Bcl-2 homologous antagonist killer (Bak). Bax and Bak subsequently act within the outer mitochondrial membrane. Permeabilization of the outer mitochondrial membrane is the pivotal event whereby signaling molecules (including cytochrome c release) gain access to the cytosol to activate common caspase end-effectors via the apoptosome and caspase-9 [91] (Figure 3).

As apoptosis is a regulated process, cell-death actions are opposed by pro-survival Bcl-2 proteins inhibiting Bax and Bak activation and this balance can be modified by cardioprotective signaling. Pharmacologic strategies inhibiting apoptotic cell death include opioids, volatile anesthetics, adenosine [92], noble gases, propofol and dexmedetomidine, as well as phosphodiesterase inhibitors [93,94]. All of these are routinely used in clinical practice. Opioids or d and k-opioid receptor activation [95,96] inhibit elevated activities of caspase-3 and -9, as well as increase the Bcl-2/Bax protein expression ratio during myocardial I/R injury [97]. Akt phosphorylation, Bcl-2, and phospho-Bad expression increased after isoflurane preconditioning, whereas Bax expression decreased [98]; findings that might not translate well into the aged heart [99]. In a porcine right ventricular infarction model xenon preconditioning was accompanied by greater numbers of caspase-3-positive cardiac myocyte compared to isoflurane during early I/R injury [100]. However, in rats the noble gas was superior to isoflurane in limiting adverse cardiac remodeling and contractile dysfunction after 28 days, whereas similar expression levels of caspase-3 were found. In the same study, myocardial infarct sizes at day 28 did not differ between controls (ketamine anesthesia) and xenon or isoflurane anesthesia [101], emphasizing the importance of prolonged experimental models in order to achieve more translatable results. Propofol also exhibited antiapoptotic effects in I/R hearts, increasing the Bcl-2/Bax expression ratio and decreasing caspase-3 activity [102]. On the other hand, the intravenous anesthetic has been shown to induce cancer cell apoptosis [103,104], indicating that its effects can be pro- and antiapoptotic depending on the cellular microenvironment. Dexmedetomidine has recently received increasing attention. The α_2_-adrenoceptor agonist activates pro-survival kinases [105] and inhibits apoptosis via the PI3K/Akt pathway [106], as well as via downregulation of hypoxia-induced factor (HIF)-1α [107]. In neonatal rat cardiac myocytes dexmedetomidine dose-dependently increased the mitochondrial membrane potential, as well as the Bcl-2/Bax protein expression ratio [108]. Further results in rats indicate anti-apoptotic effects of dexmedetomidine via inhibition of high mobility group box 1 (HMGB1) expression linked to cholinergic anti-inflammatory actions during I/R injury. HMGB1 connects the cholinergic pathway with the pro-inflammatory cascade and concomitantly reduced IL-6 and TNF-α production could alleviate various forms of cell death via modified death-receptor signaling. These findings were related to an improved echocardiographic function after 4 weeks [109].

### 5.3. Pyroptosis

Pyroptosis is linked to inflammation and activation of nuclear factor-κB (NF-κB) via damage associated molecular patterns (DAMPs), TNF-α, and interleukin (IL)-1β. This leads to the assembly and upregulation of NOD-, LRR-, and pyrin domain-containing protein 3 (NLRP3) inflammasomes in cardiac fibroblasts after 6–12 h and up to days of reperfusion. The NLRP3 inflammasome is part of the innate immune response and release of oxidized mtDNA fragments from damaged mitochondria additionally links it to apoptosis [110]. Subsequent to NLRP3 inflammasome activation, caspase-1 cleaves gasdermin D and pro-IL-1β into gasdermin D-NT and IL-1β. In consequence gasdermin D oligomers locate to the plasma membrane forming pores through which IL-1β leaves the cell further triggering inflammatory cell infiltration and cytokine production. A detailed review can be found in [111]. Pyroptosis promotes adverse cardiac remodeling [112,113] and pharmacological inhibition of NLRP3 inflammasome assembly decreased myocardial I/R injury in the murine heart [114]. In pigs, NLRP3-inflammasome inhibition reduced myocardial neutrophil influx and IL-1β levels accompanied by a reduced infarct size and a 10% improvement in left ventricular ejection fraction after 7 days [115]. Interestingly, the NLRP3-inflammasome has also been implicated as part of the cardioprotective response. The complete absence of NLRP3 in NLRP3^−/−^ mice increased myocardial infarct size and abolished cardiac preconditioning with a toll-like receptor-2 agonist via the absence of Akt phosphorylation [116]. Moreover, ischemic preconditioning was not possible in hearts isolated from NLRP3^−/−^ mice [117]. It is important to note that the latter studies only investigated acute ischemic injury, whereas the main effects of pyroptosis are more likely to come into play within later timepoints up to days after the ischemic insult and thus might set the tone for the outcome of myocardial remodeling and heart failure development.

## 6. Mitochondrial Dynamics

Mitochondria contain their own mechanisms of fusion and fission (division), whereas these mitochondrial dynamics act as quality control in cellular ageing, release of pro-apoptotic factors, and mitophagy [118]. Mitochondrial fission leads to the division of a single mitochondrion into two mitochondria in a coordinated process with the help of fission proteins. Mitochondrial fusion is the contrary coordinated by fusion proteins. Moreover, fusion proteins exert additional non-fusion actions. Fusion protein optic atrophy 1 (OPA1) alleviates mitochondrial respiration via the stabilization of mitochondrial cristae, favoring ATP synthase oligomerization [119]. These non-fusion actions of both OPA1 and the fusion protein Mfn2 seem to contribute to cardioprotection, whereas the exact role of mitochondrial fusion in cardioprotection remains unclear [120]. Myocardial I/R injury induces mitochondrial fission and pharmacological inhibition of the fission protein dynamin-related protein 1 (Drp1) with mitochondrial division inhibitor 1 (mdivi-1) reduced myocardial infarct size in the murine heart [121]. Furthermore, the administration of mdivi-1 exerted cardioprotection at the beginning of reperfusion, i.e., postconditioning [122]. In neonatal rat cardiac myocytes sevoflurane induced postconditioning was restored under “diabetic” conditions via concomitant mdivi-1 preconditioning [123]. Nevertheless, translation into humans may prove difficult as no beneficial effects of mdivi-1 were found in a large animal postconditioning model [124] and chronic or prolonged inhibition of mitochondrial fission leads to aggravated I/R injury and cardiomyopathy [125].

Mitochondrial fission is inseparably connected to mitophagy as a quality control instrument. A “classic” pathway leading to mitophagy is canonical PINK-Parkin signaling. High levels of damage-generated reactive oxygen species (ROS) and ATP depletion-mediated AMPK activation suppress mTOR subsequently priming damaged mitochondria to Parkin dependent or Parkin independent cleavage. In Parkin dependent mitophagy, PINK1 acts as the upstream regulator. PINK1 identifies defective mitochondria, whereas Parkin translocates to the mitochondrial surface and ubiquitylates numerous outer mitochondrial membrane (OMM) proteins. The ubiquitylated proteins recruit other proteins to initiate mitophagy [126,127]. Defective and tagged mitochondria are subsequently engulfed in double-membraned autophagosomes that fuse with lysosomes allowing hydrolytic degradation [128]. Myocardial I/R injury leads to pronounced mitochondrial damage and it is not surprising that these quality control mechanisms are activated and indispensable to maintain mitochondrial homeostasis. Pharmacological preconditioning with simvastatin was shown to trigger mitophagy via the PINK-Parkin pathway whereas the activation of Parkin was indispensable for cardioprotection [129]. Furthermore, although Parkin^−/−^ mice display normal cardiac and mitochondrial function, they are characterized by disorganized mitochondrial networks and significantly smaller mitochondria. Adaptation to stress is impeded and they are much more prone to myocardial infarction when compared to wild type mice [130]. Moreover, myocardial infarct size was increased in PINK1^−/−^ hearts compared to PINK1^+/+^ hearts [131]. Besides the PINK-Parkin pathway a novel mitophagy unc-51 like autophagy activating kinase (Ulk1)/ras-related protein 9 (Rab9)/RIP1/dynamin related protein 1(Drp1) pathway has recently received increasing attention during myocardial I/R injury. This alternate pathway mediates mitophagy via Rab9-associated autophagosomes and knocking out Rab9 exacerbated myocardial ischemic injury [132].

## 7. Mitochondrial Mechanisms of Regeneration

Mitochondrial biogenesis is an integral part of mitochondrial homeostasis as a process of self-regeneration in which new mitochondria are generated from those existing. Mitochondrial biogenesis can increase mitochondrial mass and helps to adapt to cellular energy demands. Mitochondrial biogenesis is mainly regulated on a transcriptional level via peroxisome proliferator-activated receptor co-activator 1 alpha (PGC-1α) initiating nuclear transcription factors such as the nuclear respiratory factor-1(NRF-1), NRF-2, and estrogen-related receptor-α (ERR-α). Transcription of both nuclear-encoded and mitochondria encoded proteins is augmented, whereas an increased expression of transcription factor A (TFAM) leads to the novel production of proteins encoded by mtDNA [133]. Mitochondrial biogenesis increases the capacity for oxidative phosphorylation, decreases oxidative stress and helps to alleviate mitochondrial dysfunction. The β_2_ adrenoceptor agonist formoterol was shown to activate the Gβγ-Akt-eNOS pathway, subsequently increasing PGC-1α mRNA expression. This is interesting, as activation of the β_2_ adrenoceptor has been shown to elicit cardiac conditioning as well [134,135]. In a cell model of hypoxia/reoxygenation acetylcholine exerted protective effects via phosphorylation of AMP-activated protein kinase (AMPK) and downstream activation of PGC-1α, enhancing ATP synthesis, mitochondrial membrane potential, and activities of mitochondrial complexes. These findings were associated with improved cell viability [136]. Similar results could be shown for melatonin via AMPK-PGC-1α-sirutin 3 (SIRT3) signaling [137]. SIRT3 is a NAD^+^-dependent protein deacetylase participating in the control of energy demand during stress conditions through the deacetylation and acetylation of mitochondrial enzymes [138]. However, the role of mitochondrial biogenesis in acute pharmacologic protection against I/R injury has not been comprehensively characterized. It is likely that a bigger role for mitochondrial biogenesis might lie in the initiation and progression of myocardial remodeling, as well as preventive measures invigorating the heart against subsequent ischemic insults, such as exercise training [139].

## 8. Conclusions, Clinical Translation, and Future Perspectives

Research on cardioprotective strategies and molecular signaling has provided numerous insights into myocardial function as well as mitochondrial mechanisms of cell death and survival. It has become clear that an interacting network spanning all cellular functions and constituents confers increased intrinsic resistance against ischemic injury. Single ‘golden bullet’ end-effectors cannot be identified, whereas a number of drugs already utilized in daily clinical practice are very well capable of eliciting this complex cardioprotective response. Mitochondria are the final frontier at the crossroads of cell death and survival, in particular via the preservation of mitochondrial respiratory chain functionality and aerobic ATP-production. In this regard, respiratory chain dysfunction is linked to the opening of the mPTP and the subsequent initiation of necrotic or apoptotic cell death. The modulation and salvation of mitochondrial respiratory function could be useful in limiting inflammation induced necrotic and apoptotic cell death in the acute and subacute phase of I/R injury. The latter particularly exerting beneficial effects onto myocardial remodeling. Selected studies on pivotal mitochondrial mechanisms of myocardial I/R injury and cardiac conditioning are summed up in Table 1.

The challenge of translating cardiac conditioning into improved clinical outcomes remains unresolved. Small clinical trials showed positive results with volatile anesthetics with decreased troponin I [140,141], as well as brain natriuretic peptide [142] release in coronary artery bypass surgery (CABG). A meta-analysis of six trials using sevoflurane preconditioning corroborated a reduction in postoperative troponin levels after on-pump CABG [143]. In 414 patients, the use of sevoflurane significantly reduced one-year mortality compared to propofol anesthesia [144]. However, in a meta-analysis including fifteen trials with a total of 1155 study patients, beneficial effects were attenuated when combining isoflurane anesthesia and RIPC and major prospective randomized controlled trials with RIPC have been disappointing. Both the RIPHeart trial [6] and the ERICCA study [145] did not show survival benefits among patients undergoing cardiac surgery in 1385 or 1612 patients, respectively. While both studies were criticized for aspects of patient selection and study design, preceding limitations at the preclinical level concomitantly fail to close the gap between bench and bedside. Preclinical studies almost exclusively utilize healthy young animals without co-medications, whereas the clinical population mainly consists of comorbid older patients. For example, it is known that arterial hypertension, advanced age [146], or diabetes mellitus abolish or dampen the conditioning response [147]. Nevertheless, an in-depth analysis of the issues and challenges relating to clinical translation is beyond the scope of the current review. An excellent analysis and suggestions for future research can be found in [148,149].

As translation of the acute cardioprotective response into clinical practice has been cumbersome and, as many drugs used in daily routine supposably already confer cardioprotection, additional benefits from single-use pharmacologic myocardial conditioning are likely futile. Moreover, preclinical studies utilizing older and comorbid animals are necessary, as well as clinical phase II studies elucidating optimal dose and timing. Moreover, patient populations beyond CABG surgery need to be elucidated. Hence, cardioprotective research needs to find ways to close the gap between bench and bedside. Furthermore, the field needs to find ways to sustain the cardioprotective response during myocardial remodeling and prohibit dysfunctional remodeling within weeks or months after I/R injury. The respective long-term conditioning protocols, molecular and pharmacologic targets preserving mitochondrial respiration, as well as modifying cell death mechanisms need to be elucidated. In this regard, it is important to remember that the goal not only consists of improved long-term survival, but also increased quality of life through ameliorated cardiac function.

## Figures and Tables

**Figure 1 ijms-22-03224-f001:**
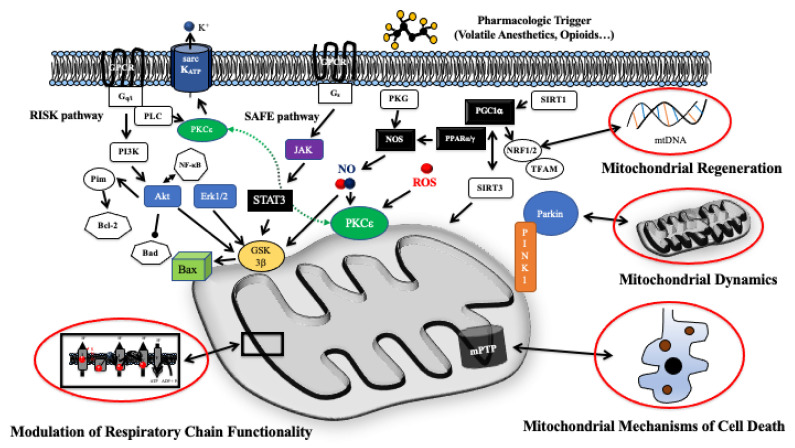
Schematic depiction of (pre-) mitochondrial signaling in myocardial I/R injury and pharmacologic cardioprotection. Mitochondria host diverse signaling modules at the crossroads of cell survival and cell death. Cytosolic signaling pathways transduce the protective signaling into resulting in modulation of mitochondrial respiratory chain activity and finally the prevention of mitochondria related cell death. Moreover, mitochondrial dynamics and mitochondrial regeneration via transcriptional changes of the mitochondrial proteome modulation contribute to an enhanced intrinsic resistance against I/R injury. Abbreviations: Akt = protein kinase B; Bax = Bcl-2-associated X protein; Bad = Bcl-2-Antagonist of Cell Death; Bcl-2 = B-cell lymphoma-2 protein; Erk1/2 = extracellular regulated kinase 1 and 2; GSK3β = glycogen synthase kinase 3β; Jak = janus-activated kinase; mPTP = mitochondrial permeability transition pore; NO = nitric oxide; NOS = nitric oxide synthase; PI3K = phosphoinositid-3-kinase; PGC1α = peroxisome proliferator-activated receptor gamma coactivator 1-α; Pim = proto-oncogene serine/threonine-protein kinase; PINK1 = PTEN-induced kinase 1; PKCε = protein kinase Cε; PKG = proteinkinase G; PPARα/β = peroxisome proliferator activated receptor α/β; NfκB = nuclear factor κB; NRF1/2 = Nuclear respiratory factor 1/2; RISK = reperfusion injury salvage kinase; sarcK_ATP_ = sarcolemmal ATP-dependent potassium channel; SAFE = survival activating factor enhancement; SIRT1/2 = Sirtuin-1/2; STAT3 = signal transducer and activator of transcription 3; TFAM = mitochondrial transcription factor A.

**Figure 2 ijms-22-03224-f002:**
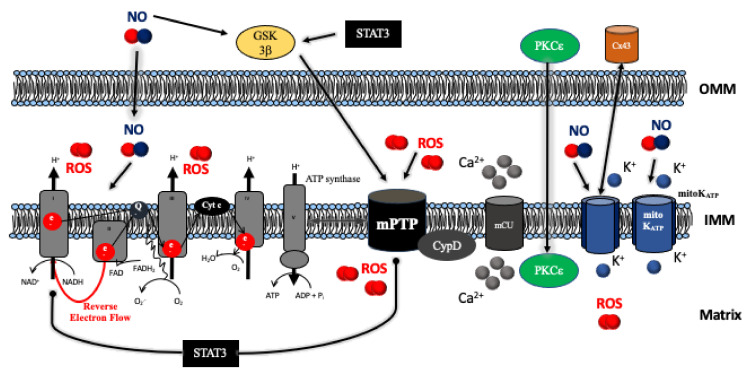
Molecular mechanisms of ischemia/reperfusion injury and cardioprotection acting onto the mitochondria. Diverse intracellular signaling pathways including the RISK (Erk1/2, Akt) and SAFE pathways (Jak) target mitochondrial functions. The centerpiece of cardioprotection is the preservation of mitochondrial respiratory function verhindern massive reactive oxygen species (ROS) production, Ca^2+^ overload, and subsequent opening of the mitochondrial permeability transition pore (mPT). Abbreviations: Akt = protein kinase B; Cx43 = connexin 43; CypD = cyclophilin D; Erk1/2 = extracellular regulated kinase 1 and 2; GSK3β = glycogen synthase kinase 3β; IMM = inner mitochondrial membrane; Jak = Janus-activated kinase; OMM = outer mitochondrial membrane; NO = nitric oxide; PKCε = protein kinase Cε; RISK = reperfusion injury salvage kinase; SAFE = survival activating factor enhancement; STAT3 = signal transducer and activator of transcription 3.

**Figure 3 ijms-22-03224-f003:**
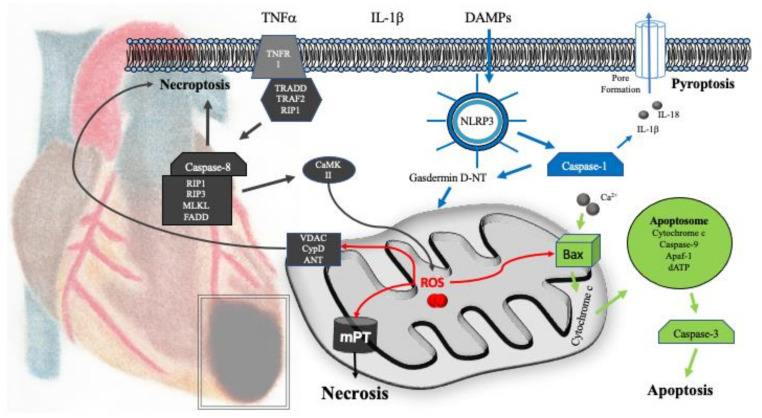
Mechanisms of cell death after myocardial I/R injury. After hypoxia, myocardial cell death is primarily driven by necrosis subsequent to the loss of mitochondrial function and opening of the mitochondrial permeability transition pore (mPTP). Necrosis also exists as a regulated form of cell death, i.e., necroptosis. Necroptosis is initiated via the activation of death receptors (e.g., tumor necrosis factor receptor 1 (TNFR1). Signaling commences via receptor-interacting protein 1 (RIP1), which mediates the activation of receptor-interacting protein 3 (RIP3) and mixed lineage kinase domain-like (MLKL) alternative activation of calcium-calmodulin-kinase II (CaMKII) by the RIP3/RIP1/MLKL/FADD complex induces mitochondrial dysfunction and membrane permeabilization via cyclophilin D, VDAC, and ANT. Pyroptosis is associated with inflammation subsequent to the ischemic insult and mediated via the NLRP3-inflammasome, caspase-1, and enhanced cytokine production. Mitochondria related apoptosis can also be elicited via mechanisms culminating in the release of cytochrome c and the formation of the apoptosome. Pharmacologic cardioprotection has been shown to act on all forms of cell death by maintaining mitochondrial function, reducing ROS production, as well as direct inhibition of apoptotic signaling or the activation of pro-survival signaling pathways, respectively. Abbreviations: ANT = adenine nucleotide translocator; Apaf-1 = apoptotic protease activating factor 1; Bax = Bcl-2-associated X protein; CaMKII = calcium-calmodulin-kinase II; DAMPs = damage-associated molecular patterns; FADD = Fas-associated protein with death domain; IL-1β = interleukin-1β; IL-18 = interleukin-18; MLKL = mixed lineage kinase domain-like; NLRP3 = NOD-, LRR-, and pyrin domain-containing protein 3; RIP1 = receptor-interacting protein 1; TNFα = tumor necrosis factor α; TNFR1 = tumor necrosis factor receptor 1; TRAF2 = tumor necrosis factor receptor-associated factor 2; TRADD = tumor necrosis factor receptor type 1-associated DEATH domain; VDAC = voltage-dependent anion-selective channel.

**Table 1 ijms-22-03224-t001:** Selected studies related to mitochondrial mechanisms of cardiac conditioning.

Authors	Model	Protocol	Identified Mechanism
Hanley, P.J. et al. 2002 [44]	Intact ventricular myocytes of guineapigs	Analyses of electron transport chain activity.	Halothane, isoflurane, and sevoflurane inhibit complex I of the electron transport chain.
Novalija, E. et al. 2003 [48]	Isolated guinea pig hearts	Sevoflurane preconditioning prior to ischemia reperfusion.	Anesthetic preconditioning preserved mitochondrial ATP production and attenuated mitochondrial ROS overload.
Baines, C.P. et al. 2005 [76]	Ppif null mice and cyclophilin D transgenic mice	Ischemia and reperfusion (24 h).	Cyclophilin D is required for Ca^2+^- and oxidative damage-induced cell death.
Krolikowski, J.G. et al. 2005 [67]	Male New Zealand white rabbits	Isoflurane pre- and postconditioning, left coronary artery occlusion and reperfusion.	Isoflurane conditioning inhibits mitochondrial permeability transition.
Chen, Q. et al. 2006 [47]	Isolated Fischer-344 rat hearts	Amobarbital preconditioning prior to global ischemia and reperfusion.	Mitochondrial damage occurs mainly during ischemia. Preserved mitochondrial respiration during reperfusion attenuates ROS release and decreases myocardial infarct size.
Ljubkovic, M. et al. 2007 [45]	Isolated rat ventricular myocytes	Analysis of mitochondrial membrane potential, redox state and oxygen consumption after isoflurane preconditioning.	Isoflurane preconditioning elicits partial mitochondrial uncoupling and reduces mitochondrial Ca^2+^ uptake.
Feng, J. et al. 2008 [43]	Isolated male adult Wistar rat hearts	Isoflurane pre- and postconditioning with global no-flow ischemia followed by reperfusion.	Identification of 26 potential phosphorylation sites in 19 mitochondrial proteins. Detection of a novel phosphorylation site in adenine nucleotide translocator-1 (ANT1).
Pravdic, D. et al. 2009 [73]	Isolated rat ventricular myocytes	In-vivo isoflurane preconditioning in the absence or presence of chelerythrine.	Isoflurane conditioning delays mPTP opening dependent on PKCε activation.
Stewart, S.; Lesnefsky, E.; Chen, Q. 2009 [53]	Isolated Fischer-344 rat hearts	Amobarbital postconditioning within global ischemia and reperfusion.	Blockade of the proximal electron transport chain at respiratory complex I attenuated maximal mitochondrial ROS generation during reperfusion.
Boengler, K. et al. 2010 [74]	Female STAT3-KO mice	Left coronary artery occlusion and reperfusion, administration of cyclosporine A prior to reperfusion.	STAT3-KO mice exhibited decreased ADP-stimulated mitochondrial respiration accompanied by increased susceptibility to mPTP opening.
Sedlic, F. et al. 2010 [46]	Isolated rat ventricular myocytes	Cardiomyocytes exposed to H_2_O_2_- after isoflurane preconditioning.	Isoflurane partially decreases mitochondrial membrane potential (ΔΨm), attenuating ROS production, decreasing Ca^2+^ uptake, and preventing mPTP opening.
Bienengraeber, M. et al. 2013 [42]	Male adult Wistar rats	Isoflurane preconditioning with left coronary artery (LCA) ligation and reperfusion.	14 mitochondrial proteins were up- or downregulated in the conditioning response, the majority belonging to complexes of the electron transport chain.
Lotz, C. et al. 2015 [49]	Male adult mice	Isoflurane preconditioning prior to ischemia and reperfusion (second window).	Isoflurane conditioning preserved the activity of respiratory complex III, stabilized mitochondrial supercomplexes III/IV, decreased malondialdehyde formation, and diminished susceptibility to Ca^2+^-induced swelling.
Szczepanek, K. et al. 2015 [57]	Transgenic mice overexpressing mitochondria-targeted, transcriptionally inactive STAT3 (MLS-STAT3E mice)	Global ischemia and reperfusion in isolated hearts. Survival analysis after left coronary artery occlusion followed reperfusion (second window).	Partial and persistent blockade of complex I in MLS-STAT3E mice decreases cardiac injury during reperfusion with concomitantly increased survival.
Zhang, G. et al. 2018 [56]	Isolated rat hearts	ZnCl_2_ postconditioning within regional ischemia and reperfusion.	ZnCl_2_ prevented ΔΨm dissipation and mitochondrial ROS generation at reperfusion by increasing mitochondrial STAT3 phosphorylation at Ser^727^ via ERK.
Lambert, J.P. et al. 2019 [17]	Tamoxifen-inducible MCUB mutant mice; MCUB knockout cell line (MCUB^−/−^)	Left coronary artery (LCA) ligation with and without reperfusion.	MCUB is incorporated into the mtCU following ischemic injury limiting mitochondrial Ca^2+^ overload and cell loss during chronic stress.
Urbani, A. et al. 2019 [62]	F-ATP synthase purified from bovine heart mitochondria	Characterization of F-ATP synthase channel activity after reconstitution and patch clamp experiments.	Ca^2+^ can transform the energy-conserving F-ATP synthase into an energy-dissipating device, i.e., the mPTP.
Lotz, C.; Stumpner, J.; Smul, T.M. 2020 [50]	Male New Zealand white rabbits	Sevoflurane compared to propofol anesthesia. Left coronary artery occlusion and reperfusion.	Sevoflurane anesthesia preserved the activities of respiratory complexes I and IV, whereas a higher portion of complex I was in its inactive (dormant) form.

## Data Availability

Not applicable.

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
