# Peer review of "Mitochondria and Pharmacologic Cardiac Conditioning—At the Heart of Ischemic Injury"

_ijms, 2021, doi:10.3390/ijms22063224_

Round 1

Reviewer 1 Report

The review by Lotz et al. is summarizing the cellular mechanisms that might be involved in cardiac conditioning. Although the topic is rather intricate and large to say the least, the authors have done an admirable work and therefore likely contribute to the complicated field of conditioning. While there are some minor omissions (lack of sarcolemmal KATP channels, very small and invisible parts of the figures, errors in abbreviations etc.), the overall review is quite well done.

My only major comment deals with the topic that is outside of the area covered by this review. The authors are leading with an impression stated throughout, that cardiac conditioning is accepted, well working and mostly beneficial in preserving cardiac and other cellular functions after ischemia/reperfusion injury. Without the doubt, a considerable experimental evidence obtained in multiple models and species has demonstrated that all forms of myocardial ischemic conditioning (pre-conditioning, per-conditioning, post-conditioning and remote preconditioning) induce very potent cardioprotection in animal models. In healthy, young hearts, many of these conditioning methods can significantly increase the heart’s resistance against ischemia and reperfusion injury.

However, essentially none of these forms of myocardial conditioning have been as effective in patients. Owing to both positive and negative results from various clinical trials, especially involving remote ischemic pre-conditioning (RIP), the results from large multicenter randomized controlled trials (such as ERICCA and RIPHeart) were meant to close that gap. It turns out that RIP, using transient arm ischemia–reperfusion, did not improve clinical outcomes in the ERICCA study, with 1,612 patients undergoing elective on-pump coronary artery bypass grafting.(26436207) Additionally, upper-limb remote ischemic preconditioning performed in 1,385 patients did not show any significant benefit among patients undergoing elective cardiac surgery.(RIPHeart 29581218) Therefore, these large multicenter trials have not only proved that ischemic conditioning was unsuccessful in cardiac surgeries; they also failed to confirm the presence of initial cardioprotection by ischemic conditioning-induced reduction of cardiac troponin release, which is a standard diagnostic indicator of myocardial injury.

Whether RIC provides more cardioprotection to the myocardium of the volatile anesthetic-anesthetized patients undergoing cardiac surgery has been examined, yet the clinical outcomes remain uncertain. Some clinical studies show that the cardioprotective effect of RIC is unable to be detected in isoflurane-anesthetized patients undergoing coronary artery bypass grafting. (PMID: 22222469, 23276595)

Therefore, I believe that some form of caution needs to be included in this review leading to the mechanisms themselves. It will be fair to say that owing to underlying risk factors that interfere with different cardioprotective interventions and the use of various drugs that interfere with cardioprotection, most of the clinical trials with cardioprotective drugs have been less than successful. The results of various ischemic conditioning in humans appear to follow the same unsuccessful path, although RIC is a potent form of endogenous cardioprotection in healthy animals. Future preclinical validation of drug targets and cardiac conditioning will need to focus more on comorbid animal models (such as age, diabetes, and hypertension) and choosing the relevant endpoints for assessing the efficacy of cardioprotective procedures to have a successful, clinical translation.

Author Response

Reviewer #1

Critique #1: Some minor omissions (lack of sarcolemmal KATP channels, very small and invisible parts of the figures, errors in abbreviations etc.)

Response: Sarcolemmal KATP channels are now included in Figure 1. We revised Figure 1 regarding the invisibility of the “Modulation of Respiratory Chain Functionality” heading, as well as Figure 2 regarding the labelling of the respiratory chain. Abbreviations have been checked for spelling errors. Furthermore, we provide high-resolution TIFF figures to the journal to allow optimal quality and readability in the final (print) version of the manuscript.

Critique #2: My only major comment deals with the topic that is outside of the area covered by this review. The authors are leading with an impression stated throughout, that cardiac conditioning is accepted, well working and mostly beneficial in preserving cardiac and other cellular functions after ischemia/reperfusion injury. However, essentially none of these forms of myocardial conditioning have been as effective in patients. Therefore, I believe that some form of caution needs to be included in this review.

Response: An additional paragraph addressing the topic of clinical translation was added to the manuscript complementing the section “Future perspectives”. This section is now named “Conclusion, Clinical Translation and Future Perspectives”, reading as follows: “The challenge of translating cardiac conditioning into improved clinical outcomes remains unresolved. Small clinical trials showed positive results with volatile anesthetics with decreased troponin I [140, 141] as well as brain natriuretic peptide [142] release in coronary artery bypass surgery (CABG). A meta-analysis of six trials using sevoflurane preconditioning corroborated a reduction in postoperative troponin levels after on-pump CABG [143]. In 414 patients the use of sevoflurane significantly reduced one-year mortality compared to propofol anesthesia [144]. However, in a meta-analysis including fifteen trials with a total of 1,155 study patients, beneficial effects were attenuated when combining isoflurane anesthesia and RIPC and major prospective randomized controlled trials with RIPC have been disappointing. Both the RIPHeart trial [6] and the ERICCA study [145] did not show survival benefits among patients undergoing cardiac surgery in 1,385 or 1,612 patients, respectively. While both studies were criticized for aspects of patient selection and study design, preceding limitations at the preclinical level concomitantly fail to close the gap between bench and bedside. Preclinical studies almost exclusively utilize healthy young animals without co-medications, whereas the clinical population mainly consists of comorbid older patients. For example, it is known that arterial hypertension, advanced age [146] or diabetes mellitus abolish or dampen the conditioning response [147]. Nevertheless, an in-depth analysis of the issues and challenges relating to clinical translation is beyond the scope of the current review. An excellent analysis and suggestions for future research can be found in [148, 149].

Reviewer 2 Report

The authors presented an interesting review discussing the mitochondrial mechanisms and pharmacologic cardiac conditioning related to myocardial ischemic injury. The manuscript is written comprehensively. Some suggestions need to consider improving the manuscript.

  1. It should be helpful to provide a table summarizing the studies on pharmacologic cardiac conditioning related to mitochondrial metabolism, at least including the studies presented in this review.
  2. The information on page 7, “lines 239—266 regarding the mPTP opening seems not to match the tile of section 4 “. Mitochondrial Respiratory Chain and ATP Production”. mPTP opening is more important in the reperfusion injury, leading to cell death, thus may fit the next section 5 better than Section 4.
  3. Adding some new progress in the clinical trial or preclinical translational studies may inspire the research interest in this field.
  4. Future perspectives need to be extended to highlight the current challenges and potential specifically in this field.

Author Response

Critique #1: It should be helpful to provide a table summarizing the studies on pharmacologic cardiac conditioning related to mitochondrial metabolism, at least including the studies presented in this review.

Response: A table summarizing prominent studies on mitochondrial mechanisms in cardiac conditioning is now included in the revised version of the manuscript (Table 1).

Critique #2: The information on page 7, “lines 239—266 regarding the mPTP opening seems not to match the title of section 4 “. Mitochondrial Respiratory Chain and ATP Production”, mPTP opening is more important in the reperfusion injury, leading to cell death, thus may fit the next section 5 better than Section 4.

Response: The respective section has been moved from section 4 to section 5.

Critique #3: Adding some new progress in the clinical trial or preclinical translational studies may inspire the research interest in this field.

Response: An additional paragraph addressing the issues of clinical trials and needs of preclinical translational studies has been added to section 8 of the manuscript.  

Critique #4: Future perspectives need to be extended to highlight the current challenges and potential specifically in this field.

Response: Please also refer to our response to Critique #3. We expanded the section “Conclusion and Future Perspectives” and renamed it to “Conclusion, Clinical Translation and Future perspectives”.

Round 2

Reviewer 2 Report

The authors could address the comments, and the manuscript has been improved.